# Effect of Sperm Cryopreservation in Farm Animals Using Nanotechnology

**DOI:** 10.3390/ani12172277

**Published:** 2022-09-02

**Authors:** Muhammad Faheem Akhtar, Qingshan Ma, Yan Li, Wenqiong Chai, Zhenwei Zhang, Liangliang Li, Changfa Wang

**Affiliations:** Research Institute of Donkey High-Efficiency Breeding and Ecological Feeding, College of Agronomy, Liaocheng University, Liaocheng 252000, China

**Keywords:** sperm cryopreservation, cryodamage, farm animals

## Abstract

**Simple Summary:**

Optimum production from farm animals can be achieved by focusing on all aspects, including reproduction in male stock. Sperm cryopreservation is a commonly practiced technique to preserve the semen of highly productive males. Semen should be collected and stored with utmost care because various factors affect their quality after collection, including the handling and cryodamage of sperm. It lowers semen quality, resulting in economic loss and low productivity. The present review paper focuses on advancements in semen cryopreservation in farm animals and methods for avoiding sperm cryodamage.

**Abstract:**

Sperm cryopreservation is one of the sublime biotechnologies for assisted reproduction. In recent decades, there has been an increasing trend in the use of preserved semen. Post-thaw semen quality and values vary among animals of the same species. Similarly, there are species-specific variations in sperm morphology, i.e., sperm head, kinetic properties, plasma membrane integrity, and freezability. Similarly, the viability of sperm varies in the female reproductive tract, i.e., from a few hours (in cattle) to several days (in chicken). Various steps of sperm cryopreservation, i.e., male health examination, semen collection, dilution, semen centrifugation, pre- and post-thaw semen quality evaluation, lack standardized methodology, that result in differences in opinions. Assisted reproductive technologies (ART), including sperm preservation, are not applied to the same extent in commercial poultry species as in mammalian species for management and economic reasons. Sperm preservation requires a reduction in physiological metabolism by extending the viable duration of the gametes. Physiologically and morphologically, spermatozoa are unique in structure and function to deliver paternal DNA and activate oocytes after fertilization. Variations in semen and sperm composition account for better handling of semen, which can aid in improved fertility. This review aims to provide an update on sperm cryopreservation in farm animals.

## 1. Introduction

In 2021, the human population increased to 7.9 billion and was expected to grow to 9.3 billion by 2050. Such an increase in population requires the exploration of better quality and cheaper protein sources from plants and animals [1]. Better farm production (meat, milk, eggs) can be achieved by focusing on all aspects, e.g., feed, fodder quality, disease control, animal husbandry practices, management and reproduction. Improving various animal reproduction technologies is inevitable to optimize high production.

The discovery of cryoprotective agents, e.g., egg yolk and glycerol, proved to be a milestone in semen quality protection during cooling and improved animal reproduction [2,3,4]. Cryopreservation is the best tool for adapting animal species to global changes (breeding accidents, epidemics) and conserving farm animal genetic resources. It also ensures the storage of endangered breeds/species genetics as well. Sperm cryopreservation produces genetically improved animals from generation to generation, and semen transports high pedigree males without distance boundaries. Farmers do not need to wait for the breeding male to inseminate female animals and can inseminate at optimal times (estrus). Semen quality varies from animal to animal and between species, as well as having higher and lower sperm cryotolerance [5]. A deluge of literature focuses on improving freezing regimes of semen for improved animal production. During sperm cryopreservation, cryodamage plays an important role in reducing semen quality. Various studies enunciate the maintenance of sperm structure and functionality during cryopreservation by optimizing the cryoprotectant’s concentration [6]. Sperm are sensitive to temperature changes during cryopreservation [7]. In this context, the quality of post-thaw semen is a burning issue that needs to be addressed. Cryopreservation changes acrosome integrity and mitochondrial activity and increases ROS production. This results in mRNA and miRNA degradation and degrades the integrity of the nucleoprotein structure. Therefore, structural and molecular alterations following sperm cryopreservation are important to discuss because each semen ejaculate contains millions of sperm. Sperm–egg fusion is critical for fertilization [8]. In the following parts of the paper, we will discuss, step by step, various structural and molecular alterations in sperm during semen cryopreservation.

## 2. Cell Membrane Changes after Sperm Cryopreservation

The addition of cryoprotectants focuses on the ability of sperm to bear temperature fluctuations without losing their quality and functions [9]. Temperature alteration, also called thermal stress, forms ice crystals inside sperm cells and the surrounding environment, which ultimately results in cryodamage [10]. Osmotic stress leads to dehydrated cells, and an imbalance in extracellular and intracellular solute contents leads to crystals forming inside pig sperm cell [11]. The extracellular medium ultimately elevates sperm volume by passive diffusion in equine [12]. During semen freezing and thawing, these factors affect lipid protein complexes, affecting the sperm plasma membrane in Chinese hamster [13]. As the temperature lowers, the phospholipid configuration disturbs as they enter laterally in the cell membrane, resulting in the adhesion of proteins. All this process weakens sperm plasmalemma and increases its permeability, ultimately lowering sperm metabolism in poultry [14].

## 3. Molecular Changes after Sperm Cryopreservation

There are major differences in poultry and bovine reproductive biology, and it affects sperm cryopreservation, e.g., average ejaculate volume of poultry ranges from 0.1–0.3 mL as compared to 5–8 mL in the bull. Poultry semen, which is higher in concentration (6–10 × 10^9^ sperm/mL) is excessively diluted [15]. The fertilization ability of poultry semen (frozen or thawed) is lowered due to dilution. The nucleoprotein structure of sperm is composed of protamine 1 (P1), protamin 2 (P2) and histones and is one of the factors in cryodamage to DNA [16]. Cryodamage is directly proportional to the type of protamines (P1 and P2) present in sperm chromatin in eutherian mammals [17]. The amount and presence of protamines (P1 and P2) vary among animal species [18]. The DNA integrity of sperm is also affected by other factors, e.g., temperature, oxidative stress due to ROS production and mechanical stress in chromatin [19,20]. Factors affecting sperm damage during cryopreservation result in the lowered fertilizing ability of frozen or thawed semen. Sperm’s messenger RNA (mRNA) can be affected by cryopreservation, which can impair its functionality [21]. Higher levels of mRNA’s AK1, IB5, TIMP, SNRPN2 and PLCz1 in frozen thawed bull semen are directly linked to higher fertility [22] In oocytes, mRNA aids in the synthesis of proteins during the initial stages of embryogenesis [23]. Transcriptionally, sperm are silent cells that cannot replace the lost mRNA during cryopreservation [24]. Freezing and thawing processes are affected by epigenetic factors. These factors affect gene expression levels, post-translational histone modifications, chromatin remodeling, DNA methylation and non-coding RNAs [25,26]. The frozen thawed semen of bull had a differential abundance of 86 microRNAs. 40 out of 86 microRNAs were responsible for sperm motility, viability, apoptotic variations and metabolism [27]. In pig fresh and frozen semen, 135 miRNAs were reported in differential abundance and 34 out of these miRNAs were involved in apoptotic alterations and various pathways of metabolism [28]. Therefore, cryopreservation alters genes related to apoptotic changes, mitochondrial membrane, DNA fragmentation, phosphatidylserine externalization and caspase activation [29,30].

## 4. Effect of Sperm Cryopreservation on Embryo

Sperm DNA methylation is an inevitable aspect of embryo development and freezing and thawing procedures affect DNA methylation [31]. This phenomenon involves the attachment of the methyl group to cystosines of CpG regions [31]. In horse sperm, DNA methylation is greater after cryopreservation (0.6% in fresh semen and 5.4% in thawed semen) [32]. Aberrant DNA methylation during cryopreservation can be responsible for fertilization failure during artificial insemination. Early embryonic development is also affected by cryopreservation. During fertilization, oocyte inherits epigenetics from nucleosomes, methylated DNA and paternal chromatin [33]. The embryonic development of the embryo is affected by transcription factors [34]. After A.I. with frozen thawed semen, downregulation of transcription factors was observed in horse embryos. [35]. Delay in embryonic development can be linked to downregulated transcription factors, i.e., TCF7L1, BTEB3, CPBP, KLF3 and NF-1 [34]. In cattle, the evaluation of fresh and frozen thawed semen showed altered mRNA and miRNA profiles after cryopreservation [30]. mRNAs’ maternal functions can be controlled by sperm-born miRNA [36]. The mRNA profiles of embryos can be altered by sperm cryopreservation because miRNA gene modulation affects some functions that negatively impact embryo development [37]. Similar alterations were observed in frozen thawed pig’s semen when glycerol was used as cryoprotectant [38].

## 5. Redox Imbalance and Mitochondria

In the cryopreservation process, an imbalance between ROS production and the antioxidant defense mechanism leads to oxidative stress. Oxygen free radicals, e.g., hydrogen peroxide (H_2_O_2_), superoxide anion (O_2_^−^) and hydroxyl radical (OH^−^), are included in ROS [39]. For normal sperm functionality, ROS are inevitable. Sperm cryopreservation activates apoptotic pathways, and ROS concentrations enhance the optimal value [40]. Enhanced ROS concentration causes structural damage to sperm’s DNA, which disrupts fertility [41]. During cryopreservation, alterations in sperm mitochondrial membrane potential occur from ROS production [29]. Static oxidative reducing potential (sORP) determines the redox balance in cryopreserved stallion semen [24]. The addition of Rosiglitazone can improve the mitochondrial membrane activity of sperm in cryopreserved stallion semen. It reduces caspase-3 activity and ultimately delays the activation of apoptotic pathways. In cryopreservation, the redox balance is also achieved by adding Rosiglitazone. In stallion semen, rosiglitazone addition aids in AKT protein phosphorylation, which balances apoptotic pathways and the survival of cells [24]. During sperm cryopreservation, ROS production is well reported in horses compared to other farm animals. Before freezing, ROS production in the mitochondria is less in pigs [42]. In cattle and buffalo, the impact of ROS on mitochondrial membrane integrity and lipid peroxidation is clearer in fresh semen than in frozen thawed semen [43].

## 6. Effect on Sperm Motility

Sperm head shape and size are not only affected by cryopreservation, but their movement is also disrupted. Cryopreservation exerts oxidative stress on sperm, causing cryodamage and reducing progressive motility in humans [44]. Cryopreservation affects the composition of sperm’s membrane lipids, survival rate, motility and damages DNA in humans [44]. Sperm motility is significantly affected after cryopreservation [44]. Computer-assisted sperm analysis (CASA) aids in evaluating sperm kinetics after freezing and thawing [45]. Morphological alterations in sperm can be observed with a phase contrast microscope camera connected to the computer screen. The trajectory movement of sperm can be recorded in continuous sequential images [46]. With the aid of CASA, sperm movement (mean), motility and velocity are measured in a time unit. Other important factors, e.g., oscillation, straightness and linearity, can also be determined. CASA also evaluates the beat frequency and lateral movement of the sperm head [47]. Using the CASA system, taking into account all variables and evaluation after applying statistical models, sperm can be classified into groups based on their motility [48]. The classification of sperm based on motility and morphology explored various aspects for understanding sperm biology. One important factor is the effect of temperature variations on sperm motility; for example, in a given semen sample, sperm motility is affected by thawing and freezing in bulls, stallions, rams and boars [46]. In some studies, sperm motility was interlinked with fertilizing capacity and viability [48].

In bulls, sperm populations having swift and non-linearity in movements have been proven to be higher in post-thaw sperm viability [49]. Cryopreservation can alter the plasma membrane, ultimately affecting sperm motility patterns. Various technological tools can be used to evaluate sperm movement, various technological tools, e.g., use of high magnification cameras and 3D technology [50,51]. Table 1 illustrates the variations in sperm cryopreservation in various farm animals. According to the available literature, various factors are progressive motility % (PM), acrosome integrity % (A.I), mitochondrial function % (MF), reactive oxygen species % (ROS), sperm motility and viability%.

## 7. Markers of Freezability

It is evident that the individuality of cryopreserved semen varies among the same species of sires and among the same ejaculates [11]. Various types of stress, including osmotic and thermal stress, are predominant during thawing and freezing procedures and affect sperm response and efficiency. Seminal plasma (SP) proteins affect sperm’s ability to cryopreserve farm animals, such as cattle, pigs, horses and sheep [57]. Sperm cryopreservation is affected by these SP proteins, as they vary from individual to individual [58]. During cryopreservation, these proteins experience carbonylation, which aids oxidation, resulting in malfunctioning and alterations in the protamine of SP proteins. These proteins resist oxidative stress and apoptotic changes [42,59]. AKAP4 and proAKAP4 proteins detect semen cryodamage in sheep, horses and pigs [60]. Higher concentrations of HSP90AA1 and HSPA8 proteins enhance the chances of sperm cryopreservation in bulls [61,62]. Higher post-thaw sperm viability and motility were observed after a higher concentration of HSP90AA1 protein in pigs. Similarly, aquaporins control the permeability of water and cryoprotectants in the cell plasma membrane during cryopreservation [63,64]. In pigs and cattle, AQP3 and AQP7 proteins positively affect sperm cryotolerance. Another important protein for sperm cryopreservation in pigs is VDAC2 [65]. In addition, a higher amount of GSTM3 in pig sperm ensures higher freezibility after cryopreservation [66]. A higher concentration of SP proteins also assists in Cryotolerance. Fibronectin-1 (FN1) aids the thawing and freezing processes in boars [67]. In sheep, the proteins TCP-1 and 26 S proteasome aid in sperm cryopreservation [68].

## 8. Role of Cryoprotectants

Preserving various tissues, cells and organs for a specific time suppresses cell metabolism. The most important factor for sperm is maintaining its functionality in an ambience that should provide inevitable conditions such as suitable pH, energy, temperature and osmolarity to limit physical damage. Tris, egg yolk and glucose are used as sperm cryopreservation media for various species. Cryoprotective media prevent ice crystal formation and cold shock that cause alteration in sperm functionality in humans [69]. Permeable and non-permeable cryoprotectants contain 4–5% glycerol and 20% egg yolk, which protect against sperm cryodamage. Egg yolk contains low-density lipoproteins (LDL) that bind plasma membrane and form a protection for sperm’s lipid bilayer [70]. Sperm post thaw quality, viability and motility greatly improved after adding LDL in place of whole egg yolk in bovine [71], swine [72], ovine [73] and equine [74]. An optimum concentration of 9% LDL lowers sperm DNA damage during cryopreservation in swine [72]. Due to bacterial contamination caused by egg yolk, researchers and scientists are searching for alternatives to substitute egg yolk as cryoprotectant substances of plant origin, potentially substituting for egg yolk [75]. Soy lecithin is a potential alternative to low-density lipoproteins (LDL), having the same action mechanism [76]. Replacing egg yolk with soy lecithin resulted in upgraded post-thaw motility (19%) in bull sperm [77], while some reports reported nil impact on motility and negative impact of soy lecithin [78]. Although some literature has proven the positive effects of soy lecithin (vegetable origin), egg yolk (animal origin) is still used as cryopreservation outcomes vary significantly. Liposomes are free from contamination and are used as an alternative to freezing media in sperm cryopreservation of equine, swine and bovine families [79,80]. The plasma membrane of liposomes controls the permeability of water and cryoprotectants [81]. For bovine sperm cryopreservation, alcohol, glycerol and ethylene glycerol are permeable cryoprotectants [81]. The presence of glycerol after thawing harms sperm, and this sensitivity varies from specie to specie [82]. A higher glycerol concentration in thawed swine sperm lowers plasma membrane fluidity [83]. A higher concentration of glycerol >3.5%—changes the properties of F-actin (which belongs to globular proteins) in the cytoskeleton [84]. A deluge of literature is available exploring alternatives, but glycerol is still considered the best cryoprotectant. Researchers have tried to blend lower concentrations of glycerol with L-glutamine and trehalose in boar semen [85]. Glycerol molecules enter lipid bilayers and cause changes in membrane diffusion rates of electrolytes, such as Na^+^ and K^+^, that result in osmotic contraction of sperm cells so that sperm can tolerate low temperatures, i.e., −79 °C. 1% glycerol and 250 mM trehalose markedly elevated acrosome integrity and motility compared to glycerol (4%) in swine [86]. Mixed trehalose (60–100 mM) and miner concentration of glycerol (1−3%) improved the quality of ovine’s frozen sperm DNA as compared to glycerol (5%) alone [87]. On the other hand, a combination of glycerol and trehalose did not affect bovine sperm quality [88]. Amides are another class of cryoprotectants. Amides cause less damage to sperm during cryopreservation due to their lower molecular mass in stallion [82]. As sperm, resilience varies among species, so amides behave accordingly in a freezing medium. Replacing glycerol with dimethylformamide and methylformamamide enhanced horse sperm acrosome integrity, mitochondrial membrane potential, motility and viability [89]. In cattle sperm, a combination of 4.7% methyl formamide and 2.3% glycerol improved the integrity and motility of the plasma membrane [90].

## 9. Addition of Seminal Plasma Components and Other Supplements

During semen cryopreservation processing, seminal plasma is removed and substituted with freezing media. Complete or partial removal of seminal plasma lowers sperm cryotolerance and freezing capacity in rams [91]. It is recommended that seminal plasma components be included to improve sperm quality. Inclusion of 5% of seminal plasma afore freezing raised plasma membrane integrity (10.5%) and motility (up to 9.2%) in frozen-thawed swine sperm [92]. Including seminal plasma in sperm reduces the effect of cryodamage by decreasing the temperature. Inclusion of 20% seminal plasma improved motility (14.7%), plasma membrane integrity (10.4%) and chromatin decondensation (13.9%) in ovine sperm [93]. The addition of seminal plasma proteins to freezing media has vital functions, from ejaculation to fertilization (in the female reproductive tract) between sperm and oocyte ovine [94]. In boar sperm, a family of proteins called spermadhesins tends to attach to glycoproteins inside the oviduct, improving sperm membrane quality [95]. At the same time, the function of spermadhesins is merely elaborated in the cryopreservation of sperm. In pig sperm, including seminal plasma proteins AQN-1, AWN and AQN-3 inhibited the capacitation of sperm at 5 °C, while adding the same proteins in freezing media produced different outcomes [96]. BSPs tend to improve plasma membrane integrity and prevent capacitation [97]. Adding RSVP14 and RSVP20 (members of SP proteins) in a freezing medium improved sperm viability, motility and acrosome integrity in swine [98]. This indicates that sperm quality can be improved or at least sustained by these specific proteins during cryopreservation [99,100]. Seminal plasma proteins having a low molecular weight (14–16 kDa) help maintain bovine sperm viability in a frozen medium [101]. Adding 1–1.5 mg of protein having M.W (14–16 kDa) elevated sperm viability by up to 20%. Seminal plasma (SP) proteins of one species are used as sperm cryopreservation media for another species. Bovine SP proteins were added to the freezing medium, and research was conducted in the bovine (sheep and goat) family [102,103]. A high freezing capacity of ovine sperm was observed after substituting bovine serum albumin (BSA) with egg yolk [104,105]. The post-thaw sperm quality of the buck improved after adding 4 mg/mL BSA to a freezing medium composed of egg yolk [106].

Enzymes inside seminal plasma proteins reduce sperm oxidation, and the sperm itself has a limited capacity to produce antioxidants [107,108]. Antioxidant defensive enzymes in sperm are catalase, superoxide dismutase (SOD), glutathione reductase and glutathione peroxidase. In equids, superoxide dimutase plays a vital role in sperm cryotolerance [109]. The defensive system of these enzymes is also related to seasonal breeding in various species [110,111]. Bovine post-thaw sperm quality improved after adding antioxidants to the freezing medium. Bovine post-thaw sperm viability and motility improved after adding 100 IU/mL of superoxide dimutase (SOD) [112]. Similar results were seen in post-thaw ovine sperm after adding SOD [113]. With temperature fluctuation, non-enzymatic antioxidants also play a vital role, along with enzymatic antioxidants. Glutathione (antioxidant) combines glutathione peroxidase and glutathione reductase to maintain redox balance. Post-thaw sperm quality improved after adding glutathione to freezing media [114]. Similar results were seen in pigs [115]. Inclusion of 2–5 mM glutathione in freezing media elevated the acrosome integrity and motility of sheep frozen thawed sperm [113].

Prediction of reactive oxygen species and peroxidation can also be controlled by adding substances similar to antioxidants in freezing media [116]. Optimum levels of ROS are inevitable for oocyte sperm fertilization and acrosome reactions, but as these levels exceed, the functionality of the sperm is disrupted [117]. The addition of benzoic acid lowered post-thaw DNA damage, lipid peroxidation and elevated horse sperm motility and viability [118]. The addition of taurine improved frozen thawed ovine sperm quality [119]. Similarly, the oxidative stress of the freezing extender decreased after the inclusion of L-carnitine in sheep [120].

Adding plant-based extracts to freezing media improves the antioxidant properties of freezing media [121]. Including Moringa oleifera extract in freezing media lowered the harmful effects on frozen thawed sperm quality in sheep [122]. Similarly, ovine sperm acrosome integrity, motility and viability were improved by the addition of an aqueous solution of rosemary [123]. The addition of resveratrol improved post-thaw horse sperm motility, mitochondrial membrane potential and plasma membrane integrity [124]. The addition of resveratrol in freezing medium lowered post-thaw oxidative damage to DNA and lipid peroxidation in swine sperm. The addition of resveratrol improved the functionality of SP enzymes (glutathione peroxidase, superoxide dismutase and catalase) [125].

Temperature fluctuation and osmotic stress cause DNA damage during cryopreservation. Therefore, antifreeze proteins (AFP) are added to protect sperm from cryodamage [126]. Inclusion of AFP in freezing medium improved acrosome integrity and sperm viability in ovine sperm [127]. In cattle and pigs, the post-thaw sperm viability and acrosome integrity of sperm improved after adding AFP in freezing media [128,129]. Currently, using AFP in farm animals is expensive for sperm cryopreservation due to less commercial production [130].

The addition of plasma with higher platelet numbers in freezing media has also been reported to lessen the harmful effects of cryopreservation. The platelet group has a concentration 3–7 times higher that inhibits reactive oxygen species (ROS) effects and maintains post-thaw sperm quality [131]. The addition of 5% plasma higher in platelets improved sperm viability and motility without affecting mitochondrial membrane potential and ROS production in humans [132]. The inclusion of plasma higher in platelets in freezing medium brought higher post-thaw motility in sheep [133].

## 10. Conclusions

With advancements in biotechnology, vigorous research and findings from researchers reduced cryodamage of sperm and improved its reliability, efficiency and ultimate productivity. As cryotolerance varies among species, future research should focus on evaluating the specific proteins/factors of cryodamage. Better collaborations among academia, sperm quality labs and production units (farms) are needed to improve the sperm quality of farm animals.

## Figures and Tables

**Table 1 animals-12-02277-t001:** Sperm cryopreservation variations specific to species.

Specie	PM (%)	A.I (%)	MF (%)	ROS (%)	Sperm M & V (%)	Reference
Cattle	40	10–19	15	48	50	[52]
Pig	50	30	30	2	60	[53]
Sheep	80	50	30	1.5	30–40	[54]
Horse	70	12	35	1	30	[55]
Goat	68–73	73–81				[56]

Abbreviations: PM (%), progressive motility; A.I (%), acrosome integrity; MF (%) mitochondrial function; ROS (%) reactive oxygen species; Sperm M&V (%), sperm motility and viability.

## Data Availability

Not applicable.

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
