# Peer review of "Effect of Sperm Cryopreservation in Farm Animals Using Nanotechnology"

_animals, 2022, doi:10.3390/ani12172277_

Round 1
Reviewer 1 Report
The review article entitled “Advances in semen preservation of farm animals” provides useful information about the update on semen preservation in farm animals by focusing on various structural and molecular alterations in sperms during semen cryopreservation. The review was written well from my point of view. However, I have identified minor issues that should improve the quality of the review article and produce a revised manuscript suitable for publication:
- Line 95: “4. Fertilizing ability” The title is not expressive of the written paragraph. I suggest removing this title and transferring this paragraph to the title of “3. Molecular changes after sperm cryopreservation”.
- Table 1 is not mentioned in the text. Please explain the table and define the abbreviations in the table.
- Since you are talking about recent developments in semen preservation, I suggest adding a title on the use of nanotechnology in semen preservation.
- There are a lot of hyphens (-) in the whole manuscript please revise it.
Author Response
Kindly see response letter in attachment.

Reviewer 2 Report
The authors aim to review advances in semen cryopreservation and methods to avoid cryodamage, but the article does not accomplish this objective. The text gives a review about sperm injuries after cryopreservation and its effects on fertility and embryo development; it also gives background on cryomarkers, cryoprotectants and supplements. However, the advances in the state of the art and the methods to avoid cryodamage are not clearly stated. Therefore, the title and the objectives do not reflect the contents of the paper.
Regarding the body of the article, some observations are made below:
1. Topic 2 is about sperm structural changes, but it reviews only membrane injuries. The authors should change the subtitle, or include other sperm injuries.
2. Topic 3 is about molecular changes, and it starts with differences between bovine and poultry sperm – are these differences important for the molecular changes? If yes, the authors should explain, if not, they should remove the sentence. Then the authors talk about nucleoproteins, but they do not explain how they change after cryopreservation.
3. Topic 4 is about fertilizing ability, but the authors do not make a link between mRNAs and fertility.
4. Topic 5 is about effects on embryo development, and the first sentence states that sperm DNA methylation affects freezing and thawing procedures. It should be the opposite, that freezing and thawing procedures affects DNA methylation.
5. Topic 7 is about sperm motility and it only reviews the CASA system, with no link with cryopreservation.
6. Topic 10 (page 7, line 292) the authors say that “In recent years, the quality of sperm freezing extenders has been improved by the addition of additives of animal origin. e.g. egg yolk.” However, egg yolk has been used as a cryoprotectant for a long time (page 2, line 45).
7. Table 1 (page 4) is not mentioned in the text, and the authors should include a subtitle explaining the initials used (PM, AI, MF, …).
8. Figure 1 is not necessary; all the information is already written in the text (lines 222-225).
Author Response
Respected Reviewer,
Thanks for your valuable suggestions. Kindly see response letter in attachment. Regards

Reviewer 3 Report
Please indicate the referred species in any sentence throughout the text, where it is not clear for the reader. Some examples: lines 137-142 about Rosiglitazone’s addition – lines 180-181 …HSP90AA1 and HSPA8 enhance chances of sperm cryopreservation… - 220-221 … Researchers have tried to blend lower concentrations of glycerol with L-glutamine and trehalose… etc.
Lines 53-54: change “specie to specie” to … between species
Line 95 change the title “4. Fertilizing ability” to something more relative to the next text. Fertilizing ability addresses issues like gametes’ interaction and fertility process, but the text mentions mRNAs’ role.
When something is not well known and “affected” is used, please add negatively or positively affected (e.g lines 184-185 … proteins AQP3 and AQP7 affect sperm cryotolerance)
Table 1 – line 169: change specie to species and explain in a following the table legend, all the abbreviations (e.g PM progressive motility etc.)
Lines 241-242: Change to “At semen cryopreservation processing, seminal plasma is removed and substituted with freezing media.
Please re-phrase the following sentence “Complete seminal plasma removes sperm cryotolerance and freezing capacity.” It is difficult to understand.
Author Response
Respected Reviewer,
Thanks a lot for your valuable suggestions. Kindly see response letter in attachment. Regards

Round 2
Reviewer 2 Report
After the changes, I recommend publication.